# 'Relationships on campus are situationships': A grounded theory study of sexual relationships at a Ugandan university

Vikas Choudhry[1]¤*, Karen Odberg Petterson[1], Maria Emmelin[1], Charles Muchunguzi[2], Anette Agardh[1]

1 Social Medicine and Global Health, Dept. of Clinical Sciences, Lund University, Malmö, Sweden, 2 Institute of Interdisciplinary Training and Research (IITR), Mbarara University of Science Technology, Mbarara, Uganda

¤ Current address: Sambodhi Research and Communications Pvt Ltd., Noida, Uttar Pradesh, India
* choudhry.vikas@gmail.com

**Data Availability Statement:** The study is based on personal data for which legal restrictions for accessing the data apply. The data is deposited at the Swedish University, and access to them is

## Abstract

Understanding the complexities of sexual relationships is essential to understand the risky sexual behaviours among young people in Ugandan universities. Nine focus group discussions conducted with 31 males and 33 female students in 2014 utilising the grounded theory approach explored the role of sexual relationships in their lives. **'Relationships in campus are situationships'** emerged as the core category and referred to the variety of sexual interactions within relationships among young people in a Ugandan university. The study findings indicated that sexual interactions often follow a sexual script that undergoes transitions to negotiate various situations. The sexual scripts in these *situationships* were strongly influenced by local socio-cultural norms and global aspirations among young people. Students often discussed these sexual scripts within a wider discourse on transactional sexual relationships. The motivations for transactional sexual relationships ranged from 'fulfilling aspirations' of various kinds on the one hand to 'being forced into trading sex' to overcome socio-economic vulnerabilities. Sexual relationships were facilitated by the perception of a university as a sexualized space in which one may enjoy a period of emerging adulthood characterized by exploration in relationships, access to alcohol and prolonged delay in assuming the traditional adult roles of marriage and family. The sexual scripts at the cultural level were grounded in traditional gender roles although at the same time, were under transition during university life with the growing influence of globalization and consumerism in the Ugandan society. Young men and young women must be engaged to critically challenge the implicit assumptions about sexual interactions within various situations that may put them at risk for poor sexual health outcomes.

## Introduction

Youth is often a time of physical, psychological, emotional and economic changes. The period from the late teens to the mid-twenties that is often characterized by exploratory behaviours in

regulated by the Public Access to Information and Secrecy Act (SFS 2009:400). The research material can be accessed by anyone with a legitimate interest in it. Requests should be addressed to Lund University via registrator@lu.se.

**Funding:** The author(s) received no specific funding for this work.

**Competing interests:** The authors have declared that no competing interests exist.

love, sex, educational opportunities and worldviews, can be a period of confusion and is referred to as emerging adulthood [1]. The segment of emerging adults in this population is growing all over the world including the sub-Saharan Africa (SSA) region [2, 3].

The forms, meanings, values and motivations of sexual expressions can vary across cultures as well as within particular populations [4]. Socio-cultural constructions of masculinity and femininity constitute an overarching framework within which young people enact their sexuality [5]. The meanings attached to socially acceptable behaviors within sexual relationships affect the sexual interactions that the young people choose to have [6]. Through constant social interaction and negotiation, they adopt sexual norms that are characteristic of their social groups and individual needs [7, 8]. Sexuality research has been criticized for neglecting interpersonal, community, cultural and structural influences [9] and instead contextualizing sexuality as a set of individual behaviours [10]. This critique poses a challenge for the design and implementation of holistic youth-led sexual health programs [11].

Young people 15 to 24 are at considerable risk for HIV infection. Two out of every seven new HIV infections globally in 2019 were among young people. Numerous social and developmental factors contribute to the risk of HIV transmission among emerging adults including educational and vocational opportunities, economic disparities, gender inequality, socio-cultural norms, multiple partnership formation, power dynamics within relationships, time and characteristics of sexual initiation and biological factors such as gender and age at puberty [12].

In SSA youth are particularly at risk of HIV. More than 80% of adolescents living with HIV reside in sub-Saharan Africa with eighty-two per cent of new HIV infections among young women also being in sub-Saharan Africa [13–15]. A similar situation is found in Uganda where there is an increasing HIV prevalence and incidence in this cohort [16] with young women facing a four times higher burden of the epidemic as opposed to young men [17]. Multiple concurrent sexual relationships, inconsistent condom use and exchange of sex for money, gifts or favours–referred to as transactional sex–are additional drivers of the HIV epidemic among young people in Uganda [18–21]. In addition to increasing HIV risk behaviours, a shift in HIV prevention strategies to more testing and treatment services has been suggested as a potential factor for greater HIV transmission in Uganda [22].

Research among university students in Uganda indicates a similar pattern of risk behaviours [23–25]. A survey report on the status of young people's access to sexuality education and adolescent reproductive health services in tertiary education in Uganda also indicates that approximately 70% of all university students are sexually active, one- third of them have multiple concurrent sexual partners and rate of condom use is very low [26]. Transactional sexual relationships, often characterized by sexual coercion have been documented in Ugandan university context whereby young people, mainly adolescent and young girls, reporting involvement in such relationships for financial and economic vulnerabilities, in response to emotional stressors from partners: or as a result of implicit or explicit pressure to access consumer products or achieve social status [23, 27–30]. The overall HIV prevalence in Ugandan universities was estimated to be 1.2%, which was lower than the national prevalence although higher rates of HIV were recorded among young girls in university, similar to national trends in HIV prevalence [31].

Although there is evidence of increasing sexual activity in Ugandan universities [31], little is known about how students view their sexual relationships. Previous epidemiological research among Ugandan university students has been mainly quantitative and cross-sectional in nature. The few qualitative studies to address the various types of sexual relationships in Ugandan universities have generally been limited to understanding a) the motivations of

young women who participate in transactional sex and b) economic and power asymmetries that characterize transactional sex and often lead to unplanned sexual activities [24, 32, 33].

In order to understand the nuances and complexities of the different situations within which young people at Ugandan universities may engage in sexual risk-taking behaviours, the present qualitative study seeks to explore the role sexual relationships play in the life of Ugandan university students. Understanding how young people negotiate sexual relationships at a university in Uganda can inform intervention programmes seeking to address sexuality and safe sexual behaviours within similar populations of young people.

## Background

Uganda had five public and twenty-nine private universities at the time of this study.

Despite increasing numbers of students being enrolled, the university enrolment rates are still low as compared to international statistics. According to a report on higher education by the Ministry of Education of Uganda, cited in a paper on Uganda's tertiary educational distribution [34] three out of every 10 Ugandans made the transition from primary to secondary education, and only one out of every 20 received a tertiary education. Only four out of ten students enrolled at tertiary level were women in 2015, compared to three out of ten in 2008. In addition, Ugandans universities have played a role in favouring students from higher socioeconomic status through their biased selection procedures as well [34, 35]. A study done at Makerere University also indicated that male students have access to public sponsorship for university education while female students are more dependent upon private sponsorship and their parents' educational level for entry into higher education [36].

For most young people university life marks the beginning of an independent living, in hostels on and off campus, far from their parental homes. According to a report on higher education, there were 198,066 university students in Uganda in 2011, 44% of whom were women [37]. Most undergraduates at those schools are between the age of 20 and 24, live off campus and are supported privately [31]. The current study was carried out at Mbarara University of Science and Technology (MUST), a public university in south-western Uganda, which had an undergraduate population of 2,870 students in 2014.

## Methods

### Design

A qualitative methodology using the grounded theory approach of Corbin and Strauss [38] was employed to explore the role of sexual relationships among university students in a Ugandan university. The grounded theory approach is a systematic method of qualitative research that seeks to generate new theory to explain phenomenon [38]. Grounded theory is characterized by attributes such as the constant comparison method, which aims to iteratively develop codes, categories and themes through data analysis; data collection based on emerging categories; and theoretical sampling, which involves the identification and selection of rich data sources to explain the social phenomenon [39].

Data were collected using focus group discussions (FGDs). Focus groups were considered appropriate for our study since they enabled participants to shift from a traditional conceptualization of sexual behaviour as a product of individual decisions, to a more holistic notion of sexuality as a socially negotiated phenomenon, strongly influenced by peer norms [10]. In addition, FGDs are particularly effective at quickly eliciting the social and cultural norms of a subgroup while exploring consensus and disagreements on issues of interest within the group [40, 41]. FGDs also give researchers access to conversations that often include every-day language and local terms, and allow them to observe group dynamics [41, 42].

## Sampling of informants

To be able to understand the social context and design the sampling process, the first author (VC) had a period of engagement and discussions about the study with lecturers, students and peer educators in their native university environment. The researcher also spent some time talking to young people in the public spaces such as restaurants, nightclubs, and bars.

The objectives of the study were discussed more specifically with the peer educators at MUST Peer Project (MPP), a student led project aimed at improving life skills, reproductive health and sexuality among university students. They were regarded central for recruiting participants, since they had been involved in a student led peer education project since 2003.

The first author and primary researcher (VC) employed a purposive sampling technique to obtain maximum variation in participants in terms of gender, age, year of study, and faculty of study reflecting the diversity present among young people ages 20 to 24, at MUST. Students were approached individually with the help of MPP peer educators or contacted through the notice boards on campus. The invitation included a brief description of the purpose of the study. According to the MPP peer educators, the participants in the study's nine FGDs were able to adequately represent the profile of the university students at MUST.

## Data collection

A total of nine FGDs were held from June 2014 to July 2014. We assembled six homogenous FGDs (three with men and three with women). In order to determine whether topics were discussed differently in the presence of both sexes, three additional groups were composed of mixed-sex participants. Each of the nine FGDs consisted of six to eight students and lasted between 1.5 to 2 hours. A total of 64 undergraduates (33 females and 31 males) representing all the academic faculties and scholastic years participated in the FGDs. The composition of the FGDs is presented in Table 1 below:

All discussions were conducted in English and audio-recorded after obtaining permission from the participants. The FGDs were conducted in a quiet and private surrounding in a project house (LUMUST house- Lund University and Mbarara University project house). They were facilitated by the first author (VC) with the assistance of four MPP peer educators (3 boys and 1 girl), having one MPP peer educator present for taking notes during each FGD. All participants were encouraged to actively participate in the discussions though efforts were made to not single out any one participant so that one would feel uncomfortable. After each FGD, an allowance between 10000 UGX ($ 4) and 25000 UGX ($ 10) was provided to each participant to cover their travel expenses.

**Table 1. Characteristics of focus group discussion (FGD) participants.**

| FGD # | Gender | Total no. of participants | Faculty of study |
|---|---|---|---|
| 1. | All male | 8 | Faculty of Medicine |
| 2. | All female | 8 | Institute of Interdisciplinary Training and Research, Faculty of Medicine, and Faculty of Science |
| 3. | All female | 8 | Institute of Management Studies, Institute of Computer Sciences, and Faculty of Science |
| 4. | Mixed sex | 6 (3 males, 3 females) | Institute of Computer Sciences |
| 5. | All male | 6 | Institute of Management Studies |
| 6. | All male | 7 | Faculty of Medicine and Faculty of Science |
| 7. | All female | 7 | Institute of Interdisciplinary Training and Research |
| 8. | Mixed sex | 7 (4 males, 3 females) | Faculty of Science and Institute of Computer Sciences |
| 9. | Mixed sex | 7 (3 males, 4 females) | Faculty of Science and Faculty of Medicine |
| Total | | 64 (31 males, 33 females) | |

The semi-structured topic guide for the FGDs included open-ended questions on the kind, dynamics, and motivations of various sexual relationships that students engage in, along with risk perceptions associated with those relationships. Prior to the data collection, two informal FGDs were conducted to ensure the adequacy of the topics to be discussed and the FGD technique. The moderator (VC) conducted the focus group as a discussion of sexual relationships on campus in general, enabling students to respond in the third person to avoid feeling that they were being interrogated about their own sexual behaviour.

## Data analysis

The audio recordings were transcribed verbatim by the first author (VC–one each of male, female, and mixed-sex) and a professional transcriber (remaining six FGDs). All the transcriptions were crosschecked with the audio recordings by the first author (VC) for any discrepancies. The analysis of the transcripts was guided by the grounded theory approach of Corbin and Strauss (2008), with an ideal schedule of analysing each interview prior to the next interview. We did not strictly adhere to the analysis of each FGD before the subsequent FGDs. Instead, initial categories were developed using a reflexive research journal, which was maintained by the first author (VC) after listening to the audio recording within a day of conducting the FGD. The reflexive journal contained predominant categories that emerged during each FGD which were also discussed with some of the members of the research team (KOP, ME and AA). As categories began to solidify, the primary researcher (VC) focused on some ideas more specifically and followed these up with lines of inquiry with further probing in the subsequent FGDs as per the grounded theory approach.

Finally, FGD transcripts were open coded line-by-line using participant´s language to label codes that helped us to stay close to the data [42]. The coding process was led by the first author and supervised by senior member of the research team (AA) throughout the analysis. The codes of the first two transcripts along with the coding process were then discussed in a workshop between the members of the research team (VC, AA, KOP and ME) and all the inputs were then incorporated in the subsequent coding process. Coding sought to categorize the text according to the type of sexual relationships and was done separately by gender to identify differences and similarities in values and beliefs among young men and women. The codes were compared and grouped together into related categories (axial coding) based on predominant issues that were raised across FGDs [42] as illustrated in Table 2 below.

**Table 2. The analytical process.**

| Text | Codes | Example of a sub-category | Category (built with other sub-categories) |
|---|---|---|---|
| "Most of the girls in the campus want more. And in that process you cannot get from one person because you need to get benefits all over, in all fields. So with campus students (boys) you cannot get financial benefits though you can get help in academics. . .. but you need some older man outside there who is ready to take the responsibility of your being in the university who can provide you for your finance" | 1. Girls wanting more | Covering bases for all situations | Accepting sexual relations as trading currency |
| | 2. Not getting everything from one partner | | |
| | 3. Maximizing benefits | | |
| | 4. Getting more partners | | |
| | 5. Campus boy for academic assistance | | |
| | 6. Campus boys being broke | | |
| | 7. Girls needing financial assistance | | |
| | 8. Older man taking responsibility | | |

Discussions were held among the research team to identify and explore a core category using the constant comparison method of going back and forth between codes, sub-categories and categories. The validity of the study was further enriched by analysing memos that were maintained during the data collection phase and earliest data analysis phase. Based on the analysis, representative quotations were selected to support the analysis.

### Ethical considerations

The Institutional Review Committee at MUST (IRC-MUST) granted the permission to conduct the study. The Uganda National Council of Science and Technology also approved the study design. Before the start of the FGDs consent forms were distributed to the participants, who were assured confidentiality and anonymity. Participants were informed they could withdraw from the study at any time. They were also informed that they could request the moderator to switch off the audio recorder at any point. All names and identification of participants were erased from the transcripts, which along with the audio-recordings were stored as a password protected file accessible only to two of the responsible researchers (VC, AA). Participants were informed that counselling services were available, should they become psychologically distressed during the course of the discussions.

## Findings

The findings of our study explore the role of sexual relationships in the lives of young university students in Uganda. The findings are structured around the Sexual Scripting Theory (SST) developed by Simon and Gagnon [43] that is based on the premise that sexuality is a social interaction and is grounded in the symbolic interactionist and constructivist perspectives of sexuality [44]. The authors of SST argue that sexual behaviours are constructed from the interplay among three levels of scripting: 1) cultural, 2) interpersonal, and 3) intra-psychic. *Cultural scripts* are guidelines that represent a social understanding of the collective meanings of sexual behaviours; *interpersonal scripts* represent the negotiations and behaviors of the actors with others in specific social contexts drawn heavily upon cultural scenarios; *intra-psychic scripts* refer to personal understandings of the sexual norms which guide behaviour and enable reflection upon it [45]. The text that follows is structured with categories in bold and sub-categories in italics within single quotation marks placed under the various sexual scripts as per the SST.

### Negotiating sexual situationships: An intra-psychic script

Negotiating sexual situations refer to the personal intra-psychic scripts within SST that are indicated by our core category in our analysis, *'Relationships on campus are situationships'*, an in-vivo code (a direct quote from one of the participant). It refers to the motivations and negotiations concerning sexual relationships in varying situations that are part of campus life.

> I feel that nowadays**, all relationships on campus are situationships**–for boys on campus it is about liking a girl's eyes one day, legs the next, dress the third day; so you deal with that situation by trying to get into sex with new ones every- time you like something. And for girls it is about having a boyfriend for hair, boyfriend for nails, boyfriend for clubs, boyfriend for mobile, boyfriend for school fees, and boyfriend for food.
>
> (FGD 2, female)

Broadly, young men and women located the sexual interactions within a discourse on *'transactional sexual relationships'*. Students often discussed these *'transactional sexual*

*relationships'* along a continuum of volition that consisted of voluntary sexual relationships for '***fulfilling aspirations***' at one end, to a feeling of '***being forced into trading sex***' for education and daily subsistence at the other end. However, the participants of both the sexes clearly distinguished commercial sex from transactional sex. The partners in transactional sex were positioned as girlfriends and boyfriends. The transactional sexual relationships were often longer lasting than a typical encounter with a 'malaya' (female commercial sex worker) that was often one-off and involved paying a predetermined amount of money for sexual activity. This view of transactional relationships as the only kind on campus caused discomfort among a few female participants. They pointed to some relationships they were familiar with among students, that were based on love, commitment, and sharing resources.

## From fulfilling aspirations to being forced into trading sex: A continuum of volition

Participants consistently brought up the discourse on the role of sexual relationships in '**fulfilling aspirations'**, although their aspirations were often gender specific. Both men and women believed that masculine sexual desire, the need for regular sex and sexual vitality were driven by biological factors. Young men in the FGDs described male sexual urge as physically unrestrainable, but also attributed it's heightening to the trend of provocative dressing among campus girls and the use of their feminine sexuality to seduce campus boys.

> Once a man tastes blood, he wants more [sex]. The boys are talking about different girls on football fields and about their sex life and the number of women they have done [had sex with]. . .. Anyway men being men, get easily aroused by anything and these days the women wear short clothes. African men cannot control the urge as soon as they see hips [general laughter]. . . Sometimes these women are enticing you by inviting you to their rooms and wearing almost nothing. . . then we know it's difficult for men to control.
>
> (FGD 5, male).

Males often engaged in multiple sexual partnerships with the opposite sex, something described as *'hit and run sexual relationships'*. These relationships were characterized by casual, temporary and short-term sexual liaisons with a number of partners, often based on lust and devoid of other emotions. Higher social status, acceptance in peer circles and fulfilling masculinity roles were aspirational benefits for engaging in these relationships among young men. Women and men spoke of how campus males compete among themselves for sexual favours from females, even if it means spending their monthly allowance or all their pocket money to impress them. Likewise, female students compete to date males who are apparently well off and are taking courses such as in the faculty of medicine.

> Boys from medical faculty are often talking about hit and run relationships they have and how they are dating 3–4 girls at same time from different faculties. It is natural for boys to boast about it. . ..after all they are from medical faculty and girls are very attracted to them for being doctors.
>
> (FGD 5, male).

Among young women, the *'transactional sexual relationships'* in which there was an exchange of money, services, favours, or other goods in return for sexual relations with an older off-campus man emerged as a prominent topic of discussion in all FGDs. Females explained that their peers have multiple partners, sometimes concurrently, to satisfy their material aspirations for

such things as cash, fashionable hairdos, expensive clothing, and high-end mobile phones. Female students described various scenarios for *'transactional sexual relationships'* although relationships were more often said to involve female students and working men off campus between ages 25 and 30 years who came looking for campus girls. The female students described another cross generational transactional relationship with an older man, referred to as 'sugar daddy', who was typically the age of their father and often had a wife and children.

Both young men and young women discussed multiple concurrent relationships as strategies for '**fulfilling aspirations'.** Men often cited adding to the number of their sexual conquests while women spoke of maximizing material gains as motivations for multiple relationships. Both men and women discussed the potential for hurt, heartbreak and a sense of revenge in a campus relationship that drove students towards exploring multiple relationships.

The authority of university lecturers and their role in determining academic grades also led some female students in situations where they offer or provide sex in expectation of better grades, which several students referred to as *'sexually-transmitted marks'*. This was in contrast to some discussions about young women being '**forced into trading sex'** with lecturers at the university for *'fear of failing exams'*.

> This girl I knew went to the lecturer's room wanting to know her exam results. But he said her result had gone missing and made her come again and again. He also tried to proposition her to meet him in Hotel Acacia to talk about her thesis, and when she refused all advances and sat a retest he said to her that you do not need to reappear for exams since he already knew her result. . .. Now in this situation what girl wants to keep failing after studying so much?
>
> (FGD 2, female)

A contrasting view, especially among women, referred to the socio-economic situation of families. Many young women struggle to pay for their tuition fees or for their subsistence needs for the semester. Given such challenges, these students felt that some students were '**forced into trading sex'** in exchange for funds to cover education-related expenses, basic amenities and attain connections in social networks, referred to as *'survival sex'* in our study.

> Actually at home for us, our parents don't provide enough [money], especially for the girls or ladies. You could have come from really far away places in Uganda and now you need to settle in Mbarara. You call and keep calling your mother, asking if they have put money in your account. You need money for hand outs, you need to look good. . . you basically just have to look good and so to get comfortable you end up with this older guy who takes care of all those things.
>
> (FGD 3, female)

## Cultural scripts in transitions

The negotiations of the sexual *situationships* are often influenced by existing socio-cultural scripts that transition as young people move through university years to adulthood. The cultural script in our study is indicated through upholding traditional roles of **'viewing men as providers and women as receivers'** in a context of growing **'acceptance of sexual relations as trading currency'** during university years. However,

**'anticipating the ideal relationship'** also emerged as a *situationship* in our analysis, that is, one conforming to societal norms of marriage, financial security within relationships, and

romantic notions of love, trust, and respect and indicates the aspirations of students for such a relationship after campus years.

## Accepting sexual relationships as trading currencies

An implicit expectation of sex following material exchange was normative for both sexes. Students perceived sex as a valuable currency that can be traded in various situations and is not for free. Relationships with older men were sometimes considered essential, as indicated by a female participant referring to *'cross-generational relationships as the order of the day'*. The FGDs revealed considerable pressure to adopt and accept such sexual relationships from peers of both sexes.

> You go to the rooms of some of these girls [who have older partners] and you can see that these girls know the buttons to press for a man if he wants her body. They have a flat screen TV, woollen carpet, couch, music system, and they are wearing expensive dress. All that is being given by the older man in return for having sex with her and sex is no big deal for the campus girl. . .. These girls might tell a new girl that you don't have level [status] and then pull her in.

> (FGD 2, female)

Male participants echoed their female peers, emphasizing that transactional sex was a strategy that young girls deliberately employed to attract older men who could fulfil their material aspirations. During mixed sex FGDs, men confronted women on their use of sex for materialistic needs. Women retorted by calling gifts symbols of affection and a sign of commitment from the men.

Participants also cited academic assistance or some other favours from fellow students, referred to as 'friends with benefits', in exchange for sex as a growing trend among undergraduate students. Thus, young women cited sex as a resource or commodity that could be used to accrue material benefits from older partners and supplement limited resources. Discussions in the male FGDs often centred on the futility of providing favours to young women for free and the social acceptance of exchanging favours for sex. These sexual relationships were a means of *'covering bases for all situations'* and discussed as 'trade offs' that students, particularly girls, use for negotiating the variety of situations that they often faced or may face in university life.

> 'You see sex is like a trade off for things. . .. No relationship on campus comes without strings attached. Boys expect sex after academic help or porridge or anything at the end of semester when some girls are broke. These girls will also have more partners for everything like coursework, a man to take her to clubs, a sugar daddy, and more for other stuff.'

> (FGD 8, mixed sex, female)

Both men and women spoke of some male students receiving money, gifts, or promises of employment from older richer woman, referred to as their 'sugar mamma', although this behaviour was not as widespread as relationships with a 'sugar daddy'. All male FGDs distinguished between 'sugar mamma' relationships as either experimentation or the fulfilment of sexual desire. The other FGDs discussed the exchange of sex among these men as a means of gaining material benefits, money or securing employment opportunities through older women.

## Viewing men as providers and women as receivers

Female respondents were conscious of the differences in age and access to resources in sexual relationships with campus males as compared to older workingmen. They summed this up by saying they *'preferred working class men to boys who were broke'*. The need to continually give and receive gifts as a sign of commitment was often discussed in mixed sex FGDs.

> 'Dating a campus boy is like taking care of your younger brother. They are young, immature, and are always trying to look for some way to have sex. They find it hard to fend for them and relationships don't work if both are broke. So you need a man who has some money, can take care of your needs, and will show serious interest in the girl.'
>
> (FGD 3, female)

Male participants spoke about how difficult it was to enter into sexual relationships with campus girls. A common perception among males was that *'campus girls are high maintenance'* and that female students were obsessed with material pleasures and expensive lifestyles. Young men seem to have internalized the provider role and talked about not being able to date campus women due to their inability to fulfil this role.

> 'Sometimes you see these campus girls. . .. and you are thinking that you cannot even afford to buy her one shoe that she is wearing. She looks so fashionable that you can just hope someday you get her. . ... She won't even be covered by your yearly allowance.'
>
> (FGD 6, male)

In discussions around motivations for "sugar mama" relationships, it was mentioned that some young men use the money they receive from older women to initiate and sustain relationships with women on campus.

## Anticipating the ideal relationship

Abstaining from sexual activity as the hallmark of an enduring campus relationship was discussed among females. Males, however, contested this view as impractical or impossible. Participants characterized campus relationships as *'destined to break-up'*. In spite of this, students were hopeful for their future relationships and talked of *'waiting for love, trust, sincerity and respect with financial security'* as attributes of an **ideal relationship**. Females in particular discussed that this **anticipation of the ideal partnerships** was closely related to the long-term goal of marriage. Both sexes agreed that financial security was the responsibility of the male in the relationship and that this was the most important attribute of the ideal relationship. The ideas expressed in the FGDs conformed to normative and parental views of marriage between a younger woman and an older man who a) was employed, b) had a good financial background, c) belonged to a similar religion or tribe and d) was respected in the community. Sometimes this pursuit influenced young people to engage in multiple sexual relationships.

> 'Girls can sometimes have lots of men at the same time, both older men and on campus men. Many campus boys are not ready to commit because they want to explore and find that suitable girl to marry. So girls also have lots of partners in order to carefully choose a life partner.'
>
> (FGD 3, female)

## Adulthood redefined: An interpersonal script

Interpersonal scripts are described as variations of cultural scripts, which draw underlying form and meaning from traditional cultural and intra-psychic scripts, being adapted and reinforced by societal perceptions of specific situations such as being a student at a university. These scripts were indicated through categories that students described as time of their life when they could **'explore and experiment'** and **'let themselves loose'**.

## Exploring and experimenting during campus life

Increased mobility, a lack of parental supervision, as well as access to large, new social networks at universities has created a *'feeling of freedom'* among participants who perceived campus as a sexualized space and claimed that their parents, teachers, and peers often reinforced this perception. A common view was that university life was associated with initiation of sexual relationships, which was considered as a '*coming of age on campus'*.

> I don't know what happens to us when we come here [university]. Maybe it is how strict they were about sex at school. But everyone is talking about boyfriends in the university and a lot of sex happening here. University students have a bad reputation in society. Everyone including their parents are always thinking of the sex that they [students] are having. Also, isn't it true that we are over 18 years on campus and even by law we can have sex? We are adults!
>
> (FGD 2, female)

Friends were reported to have influenced student's decisions on sexual relationships. Participants noted that students, especially younger and newer girls, were naive and that caused them to *'succumb to peer influences'* of getting into sexual relationships with senior students and older partners.

> 'Today's society is different. If you have not had sex, other people [fellow students] think something is wrong with you. So if you sit in a group of friends discussing sex, but you have not had any, everyone will laugh but also try to get you into it.'
>
> (FGD 5, male)

Students stated that belonging to a group that had specific expectations of its members, whether it was socio-economic status, a trend in fashion, having a certain type of boyfriend (like an older man), or having many girlfriends, was an important factor for sexual *situationships* among students.

## Letting yourself loose

Going to nightclubs and drinking alcohol, particularly during weekends, was described as a way to relieve the stress of attending university. In the discussions *'drinking and sex was perceived as a heady cocktail'* that was based on alcohol-created disinhibition and/or alcohol-related expectations of sex.

> 'The week is always stressful, so what do you do on weekends? You hang out with friends, you go to parties, you get drunk, you meet new friends in Vegas and The Heat [names of nightclubs]. You drink, dance, and shout. . .. And you know that alcohol has a chemical that does something to your bodily needs [sexual desires]'

(FGD 4, mixed-sex, male)

Alcohol was clearly a factor that drove young people into unplanned sexual interactions. Although the relationship itself was not necessarily alcohol induced, in some cases, it was asserted, alcohol created a sense of expectancy for sex. Discussions on those cases seem to suggest that students who desired sex went *'clubbing to meet potential partners'*. The negative role of alcohol in relation to sexual behaviours, especially for girls, was also frequently discussed in the FGDs, as indicated in the following quotation:

'Girls become carefree and sexually available, men become sexually charged, and those under the influence of alcohol practice unsafe sex.'

(FGD 1, male)

The image of a *'drunk girl and easy sex'* was the dominant discourse between both genders. Alcohol was described as a strategy that campus males employed for initiating and sometimes forcing sexual relationships with campus females.

'Guys are all the time trying sex after getting the girl drunk. You can invite the girl to your room, spike her soda with hard liquor and then she is yours to sleep with'

(FGD 6, female)

Certain drinking establishments and nightclubs were pointed out as places often visited by older men looking for young campus women and offering them free drinks. Some young women who frequent these clubs strategize about initiating transactional sexual relationships with older men. Young men may use these places as dating venues to impress the campus females or get them drunk in order to initiate sex. The all-women FGDs discussed growing concerns toward a rise in incidents of rape among campus females and blamed drinking habits of women that make them vulnerable to sexual coercion.

## Discussion

The findings of our study explore the role of sexual relationships in the lives of young university students in Uganda. The core category of the study '***Relationships on campus are 'situationships'*** describes the negotiations and motivations for sexual interactions within relationships that young people develop. The young people in universities react to sexual situations through enacting sexual scripts that are influenced by existing socio-cultural norms, individual aspirations, and socio-economic vulnerabilities.

Our study argues that the sexual scripts among young people in Ugandan universities are fluid and undergo transitions as they negotiate a wide variety of changing circumstances, referred to as *'situationships'* in university life. Our study found that the cultural scripts for behaviours in sexual relationships that young people in Ugandan university discussed were grounded in traditional descriptions of masculine and feminine sexuality and gender roles and expectations in society at large. However, at the same time the young people at Ugandan universities enact the interpersonal and intra-psychic scripts under the growing influence of globalisation and consumerism in a changing Ugandan society. The campus often serves as a sexualized space that enables young people to explore and develop the transitional sexual scripts. The sexual scripting among undergraduates in our study can be compared to the script reported by young people elsewhere in sub-Saharan Africa [10, 44, 46]. The discussion in our

study revolves around fitting the analytical model to the theory of sexual scripts that emerge in our study as illustrated in Fig 1.

As of January 12, 2015, the urban dictionary website listed several meanings of the word situationships for young people. One sense is that of an unconventional relationship, being more of a situation that the young people negotiate for varying purposes, with examples of 'friends with benefits', 'sexually transmitted marks', 'transactional sexual relationships' among others being quoted as examples. The participants in our study seem to refer to this meaning when they use the word situationships. Sexual relationships may help attain particular desired ends in terms of **'fulfilling aspirations'** or negotiating a socio-economically vulnerable situation wherein they are being **'forced into trading sex'**. A study among women in southern Malawi [47] used the Save the Children framework called 'Continuum of Volition' to clarify the range of motivations for and dynamics in transactional relationships. Our study proposes that this framework can be particularly helpful in understanding the range of *situationships* for sexual interactions among young men and women in a university environment in Uganda.

That young women are largely driven into transactional sexual relationships because of desire for luxury, social status and gaining access to social networks agrees with previous findings in SSA [32, 33, 46]. It is also supported by several studies in SSA universities where female students exchanged sex for a variety of material benefits or better grades, a practice called as consumption sex [30, 48–50]. Although consumption sex among young women was a

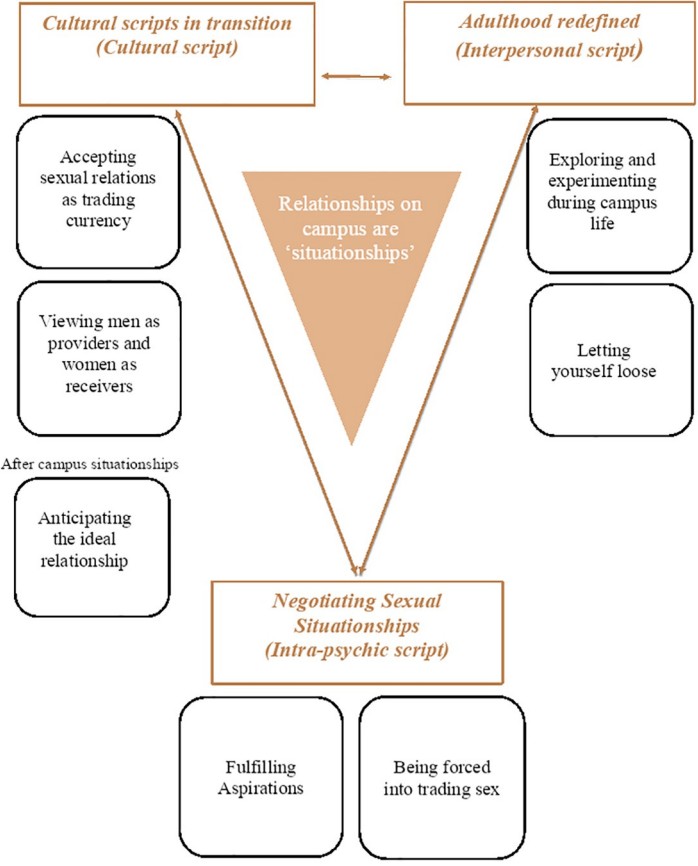

**Fig 1. Analytical model for role of sexual relationships in lives of Ugandan undergraduate students as sexual scripts.**

dominant discourse in our study, the limited resources available to such women may drive them toward engaging in survival sex. A similar finding emerged from a study done in northern Tanzania [51].

In contrast to females, a singular intra-psychic script of fulfilling sexual desires was ascribed to males who participated in sexual relationships. A study among youth in Uganda found that a socially recognized masculinity role emphasized independence from the parental home, the acquisition of money, and sexual promiscuity [52]. The researchers argue that this sexual bravado is a part of the masculinity script that was enacted through the multiple 'hit and run relationships' in our study.

The study findings also seem to suggest that young men resented women using sex for material gains, whereas women complained that (younger) men just wanted to have sex, which might suggest a misalignment between two different kinds of gendered status systems. Ultimately, the wide-spread acceptability of multiple relationships by women as well as men could be one way of aligning the two system. This may again signify the dynamic and fluid nature of sexual scripts among young people in Ugandan universities as they negotiate various *'situationships'* in their lived experiences.

Young people in universities find themselves at the cusp of adolescence and adulthood. During this developmental period, they are confronted with a campus culture that facilitates sexual relationships, but they do not yet associate themselves with the traditional adulthood roles in marriage and parenthood. Instead, they are engaged with *redefining adulthood* roles. These roles form the interpersonal scripts that construct university campus as spaces of **'exploring and experimenting'** in terms of love, sex and relationships [30, 53]. The alcohol-associated sex expectancy and disinhibition scripts, the opportunities to interact with opposite sex and the enormous peer pressures seem to create a situation of **'letting oneself loose'**, as also found in several other studies [24, 30, 53, 54]. These interpersonal scripts that result may create networks of multiple and concurrent sexual partnerships that are often transactional in nature.

Young people in Uganda appear to be increasingly affected by global popular youth culture and socio-economic processes initiated by globalization [55]. They may face the challenge of maintaining their traditions while adopting the ways of the larger globalized culture. Some studies done in SSA show that more and more young people in this region are developing bicultural identities that combine their local identity with one linked to global culture [2, 56], something we refer to as a *cultural script in transition*.

At the Ugandan university we studied, using **'sex as trading currency'** for material benefits, especially among young females, seems to be gaining acceptability. Both young men and women cited that sex as an incentive for gaining as well as giving favours or material resources. A recent study coins the term "sexual economy" of campus and argues that campus sexual economy enables students to creatively navigate existential challenges of university life [28].

The culturally acceptable scripted understanding of transactional exchange might enable young females to gain social status and accrue material benefits in a setting where there are limited means of material gain outside transactional sex. This may also provide the young girl with an ambiguous form of empowerment, a finding supported by a study in Uganda that showed young women directly associated sexual exchange with their own self-worth [57]. The socio-economic context in which young men and women in university environments live is also heavily influenced by globalization and the resulting materialistic culture. This has contributed to luxury items being increasingly considered 'needs' and a sense of *relative* poverty among young girls [4, 58, 59]. Our study finding that there is an expectation among young people in universities that a man should show appreciation of sexual relationships through sharing material resources or other forms of favours, is in agreement with another study done

among member of non-governmental organisation in southern African region in exploring contextual factors for HIV transmission [60]. Hence, young women in our study may be seen as actively strategizing to simultaneously engage multiple sexual partners by using their sexuality as a valued resource, as shown in studies on campuses in the SSA region [30, 61].

However, our study findings also indicate that young girls may not be necessarily active agents in these exchanges but may be passive victims of transactional sex fuelled by socio-economic vulnerabilities or tricked through use of alcohol. Perhaps if young women had equal access to resources as young men in terms of parental support or income generating options, they may reject transactional sex. Hence a multitude of factors like reduced access to economic resources, unmet material desires, peer pressures to own fine things, and commoditization of sex creates circumstances or *'situationships'* for transactional sex.

Our findings suggest that while transactional sexual relationships at a university may be incentivized by economic factors, the allowance for this exchange is often tied to traditional gender roles in Ugandan society that view '**men as providers and women as receivers'**. The expectation of an almost a guaranteed sexual interaction through transactional relationships may be a means of enacting masculinity among young men, characterized by a provider role and having multiple sexual partners [62, 63]. The masculine identities that largely pivot on and are socially measured by a man's ability to attract and provide resources to a woman might influence sexual relationships in young people [63]. There are varying elements of masculinity, although key requirements to attaining manhood in many African societies include achieving some level of financial independence, employment or income, and subsequently starting a family [3, 64]. The tradition of bride wealth along with growing materialism has made providing gifts and favours in exchange for sex with young girls culturally acceptable [60, 65]. A study in Malawi concluded that transactional relationships are actually part of traditional societal roles where the rich older men have an ethical obligation to share wealth on a scale appropriate to their social standing [66].

The discussions in our study show that women expect men to assume a provider role and they explained the preference for dating older men because of their better ability to provide material gifts and resources to their young girlfriends. Similarly, a study in South Africa showed that young women preferred dating and marrying men who could provide for and protect them [67]. Hence, it is not surprising that the discourse on the **'ideal relationships'** in our study denotes a typical *'cultural script'* based on traditional adult role of marriage between an older male who provides financial security and a younger female who takes care of children and family. Perhaps, this may be again indicative of a changing intra-psychic script of a *'situationship'* influenced strongly by a traditional cultural script that is often enacted after university life.

Female sexual desire was rarely spoken of during our FGDs. Both males and females referred to females as recipients of male sexual expression. The presence of a male moderator might have hindered the discussion of female sexual desire. However, in a patriarchal society like Uganda male control appears to be the norm with women expected to live up to the feminine attributes of being sexually passive and subservient to male desire thus creating unequal, coercive, and exploitative transactional relationships [30].

The economic and power asymmetries in transactional relationships, irrespective of the motivations, have been linked with sexual coercion, violence, and little or no condom use [20, 68–70]. As a result of a combination of multiple sexual partnerships with coercive and unsafe sexual practices, university students in Uganda and elsewhere are at considerable risk for HIV transmission. Policy and programmatic initiatives should consider the role of gender power differentials in transactional sex with regard to sexually transmitted infections including HIV risk. HIV interventions such as conditional cash transfers that mediate economic vulnerability

in sexual exchange should be complemented with gender-transformative programmes that target young people [71, 72]. However, these wider projects to alleviate poverty and improve gender equality will likely reduce HIV infection rate but do not challenge the normative principles of exchange in sexual relationships. Hence, young men and young women must be engaged to critically challenge the implicit assumptions about transactional sex that puts them at such grave risk.

### Methodological considerations

The paradigm for this study assumes that data are produced through an interaction between the researcher and study participants [73]. Our goal, therefore, was to reflectively interpret our data, rather than eliminate all bias introduced through the researchers' views and the research method [38].

The generalizability of these findings is limited by the sample, which was relatively small and drawn from a single, large public university in an urban setting. However, more research may be needed to examine the transferability of the findings to the Ugandan society at large and to other sub-Saharan settings, in order to avoid viewing this as a phenomenon that is restricted to students and/or elite individuals in the society.

The study also focused around heteronormative sexual relationships and these results may be less relevant to students who identify as lesbian, gay, bisexual, or transgender, or those who are questioning their sexual orientation.

Due to a period of holidays when university students were not on campus, it was difficult to recruit participants. MPP peer educators and other participants encouraged friends or acquaintances to join, introducing the possibility that they were more comfortable discussing these topics between each other than they might have been with strangers. Focus group dynamics might make it more difficult for some individuals to present conflicting views. Related to this, FGDs might provide a platform for participants to perpetuate cultural myths that are not necessarily reflected in typical individual behaviours. The researcher attempted to create a congenial and comfortable environment for the FGD with a chance for everyone to present his or her point of view without fear of feeling judged and emphasized the importance of respecting the opinions of all the participants present during FGDs. The taboo nature of sexual discussions was evident initially but tended to disappear in the course of the FGD, as seemingly the participants became comfortable with the fact that the discussions revolved around sexual relationships in general and not their individual sexual life. That the moderator came from a foreign country was somewhat advantageous: participants could assume he knew nothing about the issues beforehand, which led to open discussions. On the other hand, participants may have provided responses that they thought would please the researchers by reflecting socially acceptable context.

Although the researchers did not introduce the idea of transactional relationships into the FGDs, it is conceivable that it might have influenced data collection through extensive probing into such relationships and by hastening to identify codes and categories in data reflective of such relationships. To counter-balance such a possibility, peer-debriefing sessions were held in which a sample of transcripts were counter-coded by other researchers. The identification of relevant categories and sub-categories, and to assess the fit between the selected categories and core category for subsequent interpretation was discussed in peer debriefings. The entire research team of multidisciplinary competencies was involved in the development of the analytical model. Furthermore, for quality control, the initial results of the FGDs were cross-checked with in a mixed-sex FGD of peer-educators in MPP at the end of data collection, as a way of enhancing the credibility of the study.

## Author Contributions

**Conceptualization:** Vikas Choudhry, Karen Odberg Petterson, Maria Emmelin, Charles Muchunguzi, Anette Agardh.

**Data curation:** Vikas Choudhry.

**Formal analysis:** Vikas Choudhry, Maria Emmelin, Anette Agardh.

**Investigation:** Vikas Choudhry, Charles Muchunguzi, Anette Agardh.

**Methodology:** Vikas Choudhry, Karen Odberg Petterson, Maria Emmelin, Charles Muchunguzi, Anette Agardh.

**Project administration:** Vikas Choudhry.

**Resources:** Anette Agardh.

**Supervision:** Anette Agardh.

**Validation:** Karen Odberg Petterson, Maria Emmelin, Anette Agardh.

**Visualization:** Vikas Choudhry.

**Writing – original draft:** Vikas Choudhry.

**Writing – review & editing:** Karen Odberg Petterson, Maria Emmelin, Charles Muchunguzi, Anette Agardh.

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
