## [Decision Letter · Decision Letter 0]

28 Feb 2022

PONE-D-20-33083‘Relationships on campus are situationships’: A Grounded Theory study of sexual relationships at a Ugandan universityPLOS ONE

Dear Dr. Choudhry,

Thank you for submitting your manuscript to PLOS ONE. After careful consideration, we feel that it has merit but does not fully meet PLOS ONE’s publication criteria as it currently stands. Therefore, we invite you to submit a revised version of the manuscript that addresses the points raised during the review process.

The manuscript has been evaluated by two reviewers, and their comments are available below.

The reviewers have raised a number of concerns that need attention. They request additional information on demographic background of study participants and the research setting. They also request revisions to the presentation of the discussion; please particularly ensure that your conclusions are presented appropriately and fully supported by the results.

Could you please revise the manuscript to carefully address the concerns raised?

We look forward to receiving your revised manuscript.

Kind regards,

Marianne Clemence

Associate Editor

PLOS ONE

Journal Requirements:

Reviewers' comments:

Reviewer's Responses to Questions

**Comments to the Author**

1. Is the manuscript technically sound, and do the data support the conclusions?

Reviewer #1: Partly

Reviewer #2: Yes

2. Has the statistical analysis been performed appropriately and rigorously? 

Reviewer #1: N/A

Reviewer #2: N/A

3. Have the authors made all data underlying the findings in their manuscript fully available?

Reviewer #1: No

Reviewer #2: No

4. Is the manuscript presented in an intelligible fashion and written in standard English?

Reviewer #1: Yes

Reviewer #2: Yes

5. Review Comments to the Author

Reviewer #1: Thank you for the opportunity to review this paper. It addresses and important set of issues, is well written, and provides valuable insights. I have ten suggestions for the authors to improve their manuscript. Overall there is a lot going on in this paper and some clearer focusing of the presentation of the material would help. At present the paper feels a bit like a list of findings loosely organized around a theme, but greater clarity about that theme and organization of the data presentation would clarify its main contribution.

1.) The reader needs to know more about who goes to University in Uganda. How do they compare to the overall population? What does it cost? How do the students at this university compare to those of a similar age? What is their social position? This should not be extensive but more context is needed.

2.) I was somewhat surprised that the data were collected 8 years ago. The authors should add something about whether or not that time period was unique, or if much has changed in the interim. This need not be extensive.

3.) The reader needs to know more about selection into the sample and how it compares to the population of the university as a whole. Some of this comes at the end of the paper, but it should be in the methods section.

4.) I am somewhat concerned that the discourse in the focus groups is dealing with stereotypes rather than experiences. For example, one student is quoted saying,

And for girls it is about having a boyfriend for hair, boyfriend for nails, boyfriend for clubs, boyfriend for mobile, boyfriend for school fees, and boyfriend for food. (FGD 2, female)

This reader wondered, “is this really true? Or is it the speaker drawing a symbolic boundary between herself (who is respectable) and others, who are not. This is a significant concern. Sometimes the findings are reporting what people are doing. But at times the data seem to be respondents dismissively talking about kinds of actions they don’t like.

5.) Some claims need evidence. For example, the authors write:

“Higher social status, acceptance in peer circles, and fulfilling masculinity roles were aspirational benefits for engaging in these relationships among young men.”

However they don’t provide any evidence for this.

6.) The paper deals with two different kinds of gendered status systems that don’t align. Men want lots of partners. Women want material things. I think more time should be spent on this mis-alignment. If college men don’t have money, who are they having sex with, and why? Where are the points of alignment and misalignment within these status systems? And how are they negotiated in practice? I also return to point (4). Do all women really want material things? Or do women symbolically marked in negative ways want that? This is a distinction between rhetoric and action.

7.) The findings are presented as a kind of list, and could be organized more coherently (more on this later).

8.) The alcohol discussions are in two places and should be consolidated into one discussion.

9.) Figures 1 and 2 were not clarifying for this reader

10.) The discussion section introduces cultural, interpersonal, and intra-psychic scripts as a conceptual apparatus. I blieve that the challenges of the data presentation (which is a lot of information and at times feeling like a list) could be addressed by introducing this framework earlier, and organizing the data presentation around it. Even if the authors chose not to do this, I believe these insights must be introduced earlier, and referenced as the data are presented (and not after).

I hope these comments are useful for the authors.

Reviewer #2: Overall, I thoroughly enjoyed reading this manuscript. I especially liked the reference to urban dictionary (cite?) and support the incorporation of colloquial "emic" language and sources into academic literature. I have a few minor comments.

Introduction: Intergenerational sex is one of the major drivers of HIV in Africa, but this is not highlighted in the Introduction section although it was one of your major findings. The Background might also benefit from situating risk factors for HIV in the broader sociocultural context of Ugandan society. For example, what about gender roles in Uganda contributes? How does poverty contribute? Are university students demographically different from youth generally?

Methods:

1.The first author mentions keeping a reflexive journal. I would like more information about the authorship team and how their perspectives relate to those of Ugandan university students. Was anyone on the authorship team a university student from Uganda? How did these perspectives influence interpretation of the findings?

2. Which team members did the coding?

3. Do we have any additional detail on the demographics of the FGDs (and how these may have influenced findings)?

Overall comment, I noticed some minor typos and punctuation errors.

6. PLOS authors have the option to publish the peer review history of their article (what does this mean?). If published, this will include your full peer review and any attached files.

Reviewer #1: No

Reviewer #2: **Yes: **Irina Bergenfeld

---

## [Author Response · Author response to Decision Letter 0]

24 May 2022

Reviewer #1:

Thank you for the opportunity to review this paper. It addresses and important set of issues, is well written, and provides valuable insights. I have ten suggestions for the authors to improve their manuscript. Overall there is a lot going on in this paper and some clearer focusing of the presentation of the material would help. At present the paper feels a bit like a list of findings loosely organized around a theme, but greater clarity about that theme and organization of the data presentation would clarify its main contribution.

1.) The reader needs to know more about who goes to University in Uganda. How do they compare to the overall population? What does it cost? How do the students at this university compare to those of a similar age? What is their social position? This should not be extensive but more context is needed.

Thank you for your comment. We have added some information in our Background section (Line 112-123, Page6-7) around the dropout rates from secondary tertiary education in Uganda, higher access to tertiary education among students from higher SES and men having higher access to public scholarships:

“Despite increasing numbers of students being enrolled, the university enrolment rates are still low as compared to international statistics. According to a report on higher education by the Ministry of Education of Uganda, cited in a paper on Uganda’s tertiary educational distribution [34] three out of every 10 Ugandans made the transition from primary to secondary education, and only one out of every 20 received a tertiary education. Only four out of ten students enrolled at tertiary level were women in 2015, compared to three out of ten in 2008. In addition, Ugandans universities have played a role in favouring students from higher socioeconomic status through their biased selection procedures as well [34, 35]. A study done at Makerere University also indicated that male students have access to public sponsorship for university education while female students are more dependent upon private sponsorship and their parents’ educational level for entry into higher education [36].”

2.) I was somewhat surprised that the data were collected 8 years ago. The authors should add something about whether or not that time period was unique, or if much has changed in the interim. This need not be extensive.

Thank you again for the comment. The study was conducted as part of the PhD program for the first author VC. The context of the study remains the same and the results are still valid, as indicated by some of the other studies around transactional sexual relationships conducted in similar contexts. We have added that information in the methodological limitation as well as included some of those studies in references. 

3.) The reader needs to know more about selection into the sample and how it compares to the population of the university as a whole. Some of this comes at the end of the paper, but it should be in the methods section.

Thank you for the comment. We have included that now in the methods section (Line Line162- 169, Page 9):

“The first author and primary researcher (VC) employed a purposive sampling technique to obtain maximum variation in participants in terms of gender, age, year of study, and faculty of study reflecting the diversity present among young people ages 20 to 24, at MUST. Students were approached individually with the help of MPP peer educators or contacted through the notice boards on campus. The invitation included a brief description of the purpose of the study. According to the MPP peer educators, the participants in the study’s nine FGDs were able to adequately represent the profile of the university students at MUST.”

4.) I am somewhat concerned that the discourse in the focus groups is dealing with stereotypes rather than experiences. For example, one student is quoted saying,

And for girls it is about having a boyfriend for hair, boyfriend for nails, boyfriend for clubs, boyfriend for mobile, boyfriend for school fees, and boyfriend for food. (FGD 2, female).

This reader wondered, “is this really true? Or is it the speaker drawing a symbolic boundary between herself (who is respectable) and others, who are not. This is a significant concern. Sometimes the findings are reporting what people are doing. But at times the data seem to be respondents dismissively talking about kinds of actions they don’t like.

Thank you for your comment. The paradigm for this study assumes that data are produced through an interaction between the researcher and study participants. Our goal, therefore, was to reflectively interpret the data, rather than eliminate all bias introduced through the participant views and the research method. Hence, we have presented a very “emic” perspective on how participants described the context of all situations where university students play out their sexual relationships and behaviours. The social constructivist paradigm of the study may have influenced the data collection as well as analysis towards the conceptualisation of sexual relationships and behaviours as socially negotiated processes rather than individual decisions. Finally, the authors do acknowledge FGDs might provide a platform for participants to perpetuate cultural myths that are not necessarily reflected in typical individual behaviours. The researcher attempted to create a congenial and comfortable environment for the FGD with a chance for everyone to present his or her point of view without fear of feeling judged and emphasized the importance of respecting the opinions of all the participants present during FGDs. The taboo nature of sexual discussions was evident initially but tended to disappear during the course of the FGD, as seemingly the participants became comfortable with the fact that the discussions revolved around sexual relationships in general and not their individual sexual life. That the moderator came from a foreign country was somewhat advantageous: participants could assume he knew nothing about the issues beforehand, which led to open discussions. On the other hand, participants may have provided responses that they thought would please the researchers by reflecting socially acceptable context. We have included all the details in the relevant sections of Methodological Considerations (Page 32- 34)

5.) Some claims need evidence. For example, the authors write:

“Higher social status, acceptance in peer circles, and fulfilling masculinity roles were aspirational benefits for engaging in these relationships among young men.”

However they don’t provide any evidence for this.

Thank you for the comment. We have added a quotation in support of this in the findings section (Line 316-320, Page 17)

“Boys from medical faculty are often talking about hit and run relationships they have and how they are dating 3-4 girls at same time from different faculties. It is natural for boys to boast about it….after all they are from medical faculty and girls are very attracted to them for being doctors. (FGD 5, male).”

6.) The paper deals with two different kinds of gendered status systems that don’t align. Men want lots of partners. Women want material things. I think more time should be spent on this mis-alignment. If college men don’t have money, who are they having sex with, and why? Where are the points of alignment and misalignment within these status systems? And how are they negotiated in practice? I also return to point (4). Do all women really want material things? Or do women symbolically marked in negative ways want that? This is a distinction between rhetoric and action.

We have added some comments in the Discussion (Line 574-580, page 28) about these two different gendered status systems with regard to points of alignment and mis-alignment, but a deeper discussion of this is beyond the scope of the present study. 

“The study findings seem to suggest that men resented women using sex for material gains, whereas women complained that (younger) men just wanted to have sex, which might suggest a misalignment between two different kinds of gendered status systems. Ultimately, the wide-spread acceptability of multiple relationships by women as well as men could be one way of aligning the two system. This may again signify the dynamic and fluid nature of sexual scripts among young people in Ugandan universities as they negotiate various ‘situationships’ in their lived experiences.”

With regard to men who do not have money, possible ways to gain access to sex are by exchanging other favours (Line 393-394, Page 20), by offering alcohol (Line 502-515, Page 27), or by engaging in transactional sex with other women (Line 407-413, Page 21). 

Also, returning to point#4, our data reflect the views expressed by the participants during the group discussions, and the extent to which they might govern their individual actions is not known. This is an inherent limitation in the FGD methodology which we fully acknowledge (Line 691-694)

“Focus group dynamics might make it more difficult for some individuals to present conflicting views. Related to this, FGDs might provide a platform for participants to perpetuate cultural myths that are not necessarily reflected in typical individual behaviours.”

7.) The findings are presented as a kind of list, and could be organized more coherently (more on this later).

Thank you for your comment. We have now presented the findings in relation to the sexual scripting theory as suggested by the reviewer. We agree that introducing the theory and arranging the findings around that may present a more coherent section that is linked with the discussions in a better way for the reader. 

8.) The alcohol discussions are in two places and should be consolidated into one discussion.

Thank you for your comment. We have consolidated all the findings around role and use of alcohol among university students for sexual relationships under the category “Letting yourself Loose” (Line 491-522, Page 24-26)

Going to nightclubs and drinking alcohol, particularly during weekends, was described as a way to relieve the stress of attending university. In the discussions ‘drinking and sex was perceived as a heady cocktail’ that was based on alcohol-created disinhibition and/or alcohol-related expectations of sex. 

‘The week is always stressful, so what do you do on weekends? You hang out with friends, you go to parties, you get drunk, you meet new friends in Vegas and The Heat �names of nightclubs�. You drink, dance, and shout. . . . And you know that alcohol has a chemical that does something to your bodily needs �sexual desires�’ (FGD 4, mixed-sex, male)

Alcohol was clearly a factor that drove young people into unplanned sexual interactions. Although the relationship itself was not necessarily alcohol induced, in some cases, it was asserted, alcohol created a sense of expectancy for sex. Discussions on those cases seem to suggest that students who desired sex went ‘clubbing to meet potential partners’. The negative role of alcohol in relation to sexual behaviours, especially for girls, was also frequently discussed in the FGDs, as indicated in the following quotation: 

‘Girls become carefree and sexually available, men become sexually charged, and those under the influence of alcohol practice unsafe sex.’ (FGD 1, male)

The image of a ‘drunk girl and easy sex’ was the dominant discourse between both genders. Alcohol was described as a strategy that campus males employed for initiating and sometimes forcing sexual relationships with campus females. 

‘Guys are all the time trying sex after getting the girl drunk. You can invite the girl to your room, spike her soda with hard liquor and then she is yours to sleep with’ (FGD 6, female)

Certain drinking establishments and nightclubs were pointed out as places often visited by older men looking for young campus women and offering them free drinks. Some young women who frequent these clubs strategize about initiating transactional sexual relationships with older men. Young men may use these places as dating venues to impress the campus females or get them drunk in order to initiate sex. The all-women FGDs discussed growing concerns toward a rise in incidents of rape among campus females and blamed drinking habits of women that make them vulnerable to sexual coercion.

9.) Figures 1 and 2 were not clarifying for this reader

Thank you. We have now removed Figure 1 and only include Figure 2 (renamed Figure 1) as a combined analytical model, which we hope is easier to understand. 

10.) The discussion section introduces cultural, interpersonal, and intra-psychic scripts as a conceptual apparatus. I blieve that the challenges of the data presentation (which is a lot of information and at times feeling like a list) could be addressed by introducing this framework earlier, and organizing the data presentation around it. Even if the authors chose not to do this, I believe these insights must be introduced earlier, and referenced as the data are presented (and not after).

Thank you for this excellent suggestion. We have incorporated it in our findings section as mentioned in comment 7 above.

Reviewer #2:

Overall, I thoroughly enjoyed reading this manuscript. I especially liked the reference to urban dictionary (cite?) and support the incorporation of colloquial "emic" language and sources into academic literature. I have a few minor comments.

Thank you for your encouragement. We are really happy to hear that you picked up on the incorporation of “emic” perspectives. The authors feel that our study was designed and analysed with the express purpose of bringing to the forefront young people’s perspectives and understanding of sexual relationships in their own language and context. 

Introduction: Intergenerational sex is one of the major drivers of HIV in Africa, but this is not highlighted in the Introduction section although it was one of your major findings. The Background might also benefit from situating risk factors for HIV in the broader sociocultural context of Ugandan society. For example, what about gender roles in Uganda contributes? How does poverty contribute? Are university students demographically different from youth generally?

Thank you for your suggestion. We have incorporated literature on transactional relationships in Uganda in our introduction section (Lines 82-94 , Page 5-6):

“A survey report on the status of young people’s access to sexuality education and adolescent reproductive health services in tertiary education in Uganda also indicates that approximately 70% of all university students are sexually active, one- third of them have multiple concurrent sexual partners and rate of condom use is very low [26]. Transactional sexual relationships, often characterized by sexual coercion have been documented in Ugandan university context whereby young people, mainly adolescent and young girls, reporting involvement in such relationships for financial and economic vulnerabilities, in response to emotional stressors from partners: or as a result of implicit or explicit pressure to access consumer products or achieve social status [23, 27-30]. The overall HIV prevalence in Ugandan universities was estimated to be 1.2%, which was lower than the national prevalence although higher rates of HIV were recorded among young girls in university, similar to national trends in HIV prevalence [31].”

We have also described the differences between the university population and out of university population in Uganda in our background section (Line 112-123, Page6-7):

“Despite increasing numbers of students being enrolled, the university enrolment rates are still low as compared to international statistics. According to a report on higher education by the Ministry of Education of Uganda, cited in a paper on Uganda’s tertiary educational distribution [34] three out of every 10 Ugandans made the transition from primary to secondary education, and only one out of every 20 received a tertiary education. Only four out of ten students enrolled at tertiary level were women in 2015, compared to three out of ten in 2008. In addition, Ugandans universities have played a role in favouring students from higher socioeconomic status through their biased selection procedures as well [34, 35]. A study done at Makerere University also indicated that male students have access to public sponsorship for university education while female students are more dependent upon private sponsorship and their parents’ educational level for entry into higher education [36].”

Methods:

1.The first author mentions keeping a reflexive journal. I would like more information about the authorship team and how their perspectives relate to those of Ugandan university students. Was anyone on the authorship team a university student from Uganda? How did these perspectives influence interpretation of the findings?

Thank you for the enquiry regarding the authorship team. The first author VC is from India and was situated in Uganda during this study. The author spent a great deal of time in discussion with young people in and out of university settings, as well as with the various faculty members. Their perspectives were all incorporated in the objectives of the study, tool development, recruitment of participants, as well as data analysis. The data analysis was discussed with Must Peer Project peer educators who were students at the university. However, it was not possible to include any students in the authorship team. The authorship to them was not possible due to students moving away to employment opportunities and not being able to participate in the writing. The authorship team includes a faculty member (CM) from the university. Another author (AA) has been a PI of a peer educator project at the university and has been working with young people at the university for more than a decade. The other authors have affiliations with the university and have been part of qualitative studies that have been conducted at the university. 

2. Which team members did the coding?

Thank you for the enquiry. The coding was done by the first author VC. However, the first round of coding was also discussed in a workshop mode with authors from Lund University and then consensus was developed around coding methods and coding frame for the study. We have incorporated the details in the manuscript (Lines 212-228, Page 12):

“Finally, FGD transcripts were open coded line-by-line using participant´s language to label codes that helped us to stay close to the data (41). The coding process was led by the first author and supervised by senior member of the research team (AA) throughout the analysis. The codes of the first two transcripts along with the coding process were then discussed in a workshop between the members of the research team (VC, AA, KOP and ME) and all the inputs were then incorporated in the subsequent coding process. Coding sought to categorize the text according to the type of sexual relationships and was done separately by gender to identify differences and similarities in values and beliefs among young men and women. The codes were compared and grouped together into related categories (axial coding) based on predominant issues that were raised across FGDs (41) as illustrated in Table 2 below. Discussions were held among the research team to identify and explore a core category using the constant comparison method of going back and forth between codes, sub-categories and categories. The validity of the study was further enriched by analysing memos that were maintained during the data collection phase and earliest data analysis phase. Based on the analysis, representative quotations were selected to support the analysis.” 

3. Do we have any additional detail on the demographics of the FGDs (and how these may have influenced findings)?

Thank you for the comment. The participants of the FGDs were between 20-24 and do represent the demographic profile of university students at MUST (Line 167-169, Page 9):

“The invitation included a brief description of the purpose of the study. According to the MPP peer educators, the participants in the study’s nine FGDs were able to adequately represent the profile of the university students at MUST.”

Overall comment, I noticed some minor typos and punctuation errors.

Thank you for the comment. We have done another round of language editing and proofing.

---

## [Editor Report · Decision Letter 1]

4 Jul 2022

‘Relationships on campus are situationships’: A grounded theory study of sexual relationships at a Ugandan university

PONE-D-20-33083R1

Dear Dr. Choudhry,

We’re pleased to inform you that your manuscript has been judged scientifically suitable for publication and will be formally accepted for publication once it meets all outstanding technical requirements.

Kind regards,

Shamus Rahman Khan

Guest Editor

PLOS ONE

Additional Editor Comments (optional):

Thank you for your substantial revision to this manuscript. It has done well to respond to the two reviewers concerns and the subsequent work is much improved. No further revisions are required. However, I would encourage the authors to consider the framing of the paper around HIV. Yes, this is an important topic in SSA. However, the paper does not directly address HIV and the conclusion does not provide many helpful insights for HIV researchers. Not all papers on SSA need to be "about" HIV or framed around it. I think the HIV framing in the front could be cut back a bit (to even 1-2 sentences) and the conclusion bolstered slightly about more concrete implications of the findings for HIV research.
---

## [Editor Report · Acceptance letter]

18 Jul 2022

PONE-D-20-33083R1 

‘Relationships on campus are situationships’: A grounded theory study of sexual relationships at a Ugandan university 

Dear Dr. Choudhry:

I'm pleased to inform you that your manuscript has been deemed suitable for publication in PLOS ONE. Congratulations! Your manuscript is now with our production department. 

Kind regards, 

on behalf of

Dr. Shamus Rahman Khan 

Guest Editor

PLOS ONE